# The construction of placeness in traditional opera from the perspective of structuration theory: A case study of Huangmei Opera in Anqing, China

Fang Zhu[1]*, Qin Su[2], Jiahui Xu[1], Lei Zhu[1]

1 School of Resources and Environment, Anqing Normal University, Anqing, China, 2 School of Geography and Tourism, Anhui Normal University, Wuhu, China

* zfang315@126.com

## Abstract

Traditional opera is continuously involved in the construction of placeness because of its close connection with local culture, and the flow of the meaning of placeness has become a hot spot of cultural geography research. This study examines the driving factors and mechanisms of constructing placeness in Huangmei Opera from the perspective of structuration theory. It adopts qualitative research and positivist methodology to analyze the mutual influence and constructive relationship between Huangmei Opera and placeness. The results reveal that under the structural adjustment of the cultural system reform, multiple subjects respond to the impact of the structural adjustment through the action practices of place marketing and leisure participation, combined with the elements of placeness that the city itself possesses. The study confirms that Huangmei Opera, with its local characteristics, is a quintessential example of placeness construction and that Huangmei Opera achieves placeness construction through the interplay of structure, action, and place.

## Introduction

As a symbolic symbol of human cultural diversity and excellent traditional culture, traditional opera contains rich placeness knowledge [1,2]. In recent years, along with the need to establish cultural confidence and local cultural branding, local governments and practitioners have actively integrated local culture into cultural exhibitions and performances in order to realize the local expression of traditional opera, prompting it to become an effective part of local cultural branding, and enhancing the cultural confidence and cultural identity of residents and cultural consumers. In turn, the shaping of the local cultural brand will also promote the more active participation of stakeholder subjects in the process of constructing the placeness of traditional opera, thereby promoting the living inheritance and local development of traditional

**Data availability statement:** We recognize the importance of data sharing in ensuring research transparency and reproducibility. However, given the sensitivity of the data in this study and our ethical commitment to research participants, we have chosen to restrict data sharing in this case. The raw data from this study includes qualitative interviews, which, even after anonymization, may still contain personal narratives that could compromise participants' privacy. By the ethical protocol approved by the Ethics Committee, we guarantee the anonymity of participants and the confidentiality of the data. The contact information for the Ethics Committee at Anqing Normal University is provided below for submitting data requests. Contact information: keyanc@aqtc.edu.cn. Although we cannot share the original interview data, the methods and tools used ensure the reproducibility of the study. To this end, we have detailed the interview guidelines and thematic analysis of the interviews in the "Methodology and sample description." Additionally, we have provided a detailed interview outline in the supplementary information file and cited excerpts from the interview texts in the manuscript. This will provide other researchers with the necessary information to understand and, if needed, replicate our research methods.

**Funding:** This study was supported by the Key Project of Scientific Research in Universities of Anhui Province (2023AH050461), awarded to FZ, the Research Project of Innovative Development of Social Science in Anhui Province (2025CXZ015), awarded to FZ, and the Major Project of Scientific Research in Universities of Anhui Province (2024AH030073), awarded to LZ. There was no additional external funding received for this study.

**Competing interests:** The authors have declared that no competing interests exist.

opera. Therefore, in the geospatial practice of the two-way intertwining of globalization and localization, tradition and modernity, traditional opera with local characteristics exhibits new vitality and serves as an important carrier for maintaining placeness in the context of globalization.

The dynamic, cohesive, flexible, and interactive nature of traditional opera in cultural practice contrasts strongly with the "static nature" of tangible heritage [3,4]. Globally, traditional opera is rapidly becoming an important part of local cultural branding and tourism image construction. Based on this, geographers have begun to use traditional opera as a carrier and medium for cultural practice and place construction, exploring topics such as place identity and representation [5–8] and place perception and identity [9–12]. In this process, how economic, political, cultural, and social meanings flow through the production, exhibition, consumption, and practice of traditional opera, as well as how the meaning of placeness emerges through traditional opera, become key issues that need to be explored in depth. Therefore, it is necessary to consider the diversified subjects of placeness construction comprehensively, and based on the theoretical perspective of the mutual construction of text and placeness, to pay attention to the two-way interactive process and multiple mechanisms of traditional opera and placeness, which is an important way to seek the synergistic development of traditional opera and placeness.

Under the dual contexts of intangible cultural heritage protection and local cultural reconstruction, the study of the placeness construction of traditional opera has long faced a double dilemma at the methodological level: On the one hand, most of the existing studies remain in the static description of cultural representations and lack the dynamic examination of the triadic dialectical relationship of "structure-action-place"; on the other hand, the existing analytical frameworks tend to separate institutional constraints from the subjective practice, making it difficult to explain the temporal and spatial contextualization of the placeness construction. This paper introduces Allan Pred's structuration theory. It constructs a "three-dimensional interaction model," situating the placeness construction of Huangmei Opera within the dynamic interplay of "institutional structure," "subjective practice," and "placeness elements." The localization of this theoretical tool has a double methodological breakthrough. Firstly, from the perspective of "the time geography of structuring process," it reveals how the institutional statute facilitates place reconstruction through specific practices, such as the inheritance of class rules and the relationship between masters and disciples, in the evolution of traditional opera from grassroots troupe to national non-heritage. Secondly, it develops the analytical path of "contextualized practice" to establish a multidimensional observation system in terms of spatial configuration, subjective network, and cultural memories. The creative transformation of this theoretical paradigm not only breaks through the dichotomy of structure and action in the study of opera but also provides a new cognitive framework for understanding the modern transformation of traditional opera through the dynamic perspective of "place as a process."

Huangmei Opera has grown and flourished in Anqing, and throughout its long-term development, it has consistently maintained a close relationship with the local

community of Anqing. The placeness elements of the Anqing area provide a superior soil for the survival of Huangmei Opera, and in the continuous interweaving practice, Huangmei Opera itself also has local characteristics. Audience groups in the consumer experience, based on the place content of Huangmei Opera, produce intimate, familiar emotions; Huangmei Opera has become a medium for the transfer of placeness significance. The interaction between placeness and the development of Huangmei Opera is influenced not only by factors such as capital and power but also by the behavioral practices of the stakeholders.

This study is dedicated to revealing the dynamic inter-constructive mechanism of structure, action, and place in the construction of Huangmei Opera's placeness through the in-depth analysis of qualitative data. Based on constructivist epistemology, the study adopts a process-oriented path of qualitative analysis, focusing on capturing the dynamic relationships in three dimensions: first, how institutional structures are transformed into placeness knowledge through the strategic practices of cultural subjects; second, how the exhibition space, as a material carrier, mediates the transmission and innovation of cultural memories in daily performances; and, third, how the interactive network of multiple subjects reproduces place identities. By integrating in-depth interviews, participatory observation, and critical textual analysis, the study focuses on analyzing the "structured process" of cultural practices, specifically the interaction between institutional constraints and subjective agency within a particular spatial and temporal context. This qualitative research design aligns not only with Pride's theoretical focus on "process" and "context" but also explains the complex logic of placeness construction through cultural representations, providing a dynamic analytical framework for studying the living heritage of traditional opera.

The rest of this paper is organized as follows: Part 2 systematically comprehends the mechanism of placeness formation and the relationship between text and placeness; Part 3 explains the design logic of the qualitative research method, as well as the data sources and analysis process; Part 4 analyzes the dynamic process and mechanism of the construction of Huangmei Opera's placeness from the three dimensions of institutional structure, subjective practice strategy, and placeness elements; Part 5 discusses the findings of the study; and finally, we summarize the study's conclusions and limitations.

## Literature review

### Placeness formation mechanisms

In the 1970s, humanist scholars Yi-fu Tuan and Edward Relph introduced the concept of place into the field of geography research, and the theory of place and placeness became an important perspective guiding geographers in the study of people-place relations [13,14]. Placeness usually refers to the uniqueness of a place within a specific range of time and space [15], as well as the spirit or qualities given to a place in the process of continuous interaction between human beings and the natural environment [16]. The objective physical environment, social and cultural activities, and the emotional meaning constructed by the subject in the interaction between humans and the land are the three dimensions of placeness composition, which is characterized by the combination of objectivity and subjectivity.

The humanist and structuralist schools have been explored in the context of the concept and connotation of placeness, ultimately leading to the consideration of different perspectives on the mechanism of placeness formation. On the one hand, the endogenous mechanism of placeness formation, influenced by the humanist school of thought, is represented by the theoretical perspectives on the subjective construction and social practice of placeness. The former believes that placeness is formed through subjective construction, where different subjects construct differences in placeness. This concept is more often discussed in conjunction with the concepts of sense of place, place identity, and place attachment, among others. The latter believes that placeness is formed in the process of bottom-up action and practice by the actors.

On the other hand, placeness formation research influenced by the structuralist school focuses on the influential role of exogenous factors. Among them, scholars represented by David Harvey and Doreen Massey believe that placeness is integrated under complex social relations formed by structural forces in the course of material socio-economic processes

[17,18]; scholars represented by Michel Foucault pay attention to the role of discourse and representation in the process of placeness construction, and the resulting reconstruction of social power relations [19]. The diversified theoretical perspectives on the mechanism of placeness formation have been widely applied in empirical research, including both single-perspective studies [20,21] and comprehensive research involving humanist and structuralist perspectives [22,23], indicating that the formation of placeness is the result of the joint action of multiple forces.

Although there are differences in the perspectives that research scholars in various fields focus on, they highlight the essential feature of placeness: the uniqueness of the place [24,25]. The formation of placeness is shaped under the joint action of objective environment and subjective factors, which gives the theory of place and placeness a clear advantage in exploring the formation of people-place relations with objects as the medium.

## Relationship between text and placeness

With the continuous expansion and deepening of theoretical perspectives on the study of placeness, the new cultural geography has begun to pay attention to the interrelationship between text and placeness. Based on this, existing studies have explored how texts contribute to the construction and formation of placeness, focusing on two theoretical aspects. First is the unitary theoretical perspective. Based on the perspective of external identity, the perception and identification of the "other" group influence the presentation of placeness and its formation [26]. From a structuralist perspective, structural and non-structural dynamics are intertwined in place-making [27]. Second, the integrated perspective of humanism and structuralism. The social relations between the place and the outside world, as well as the degree to which different types of subjects identify with the placeness, drive the formation of the placeness [22]. In addition, the content records and value representations of textual placeness, as well as how different subjects symbolize and interpret it, confirm the existence of a two-way interactive relationship between the text and the construction of placeness [28–30].

The interaction between multiple influencing factors in the mechanism of placeness formation has attracted the attention of scholars, and studies from a comprehensive perspective have also shown the complexity of placeness formation, i.e., the construction of placeness is realized under the joint influence of external factors and internal mechanisms. However, in the study of the mechanism of textual participation in the construction of placeness, although attention has been paid to the joint role of humanist and structuralist factors, these two types of influencing factors are still treated as an independent existence. There is a lack of consideration of the interaction of factors from a systemic perspective.

The "structure-action-place" triple interaction proposed by Pred provides a framework for integrating structuralist and humanist research and has been applied to the field of placeness research [31]. Scholars have introduced structuration theory to explore the impact of the interactive process between social structure and the social practices of individual actors on the formation and evolution of placeness [32]. At the same time, based on the two research perspectives of social construction and humanism, they focus on the influence of macro social structure elements and individual actors' practices on the construction of placeness [33]. It has been found that the subject's "action" plays a crucial role in the process of constructing and reconstructing placeness [34,35].

The interactive relationship between structure, action, and place indicates that the formation of placeness is a practical response based on the foundation of the placeness elements by each subject of action under the structured adjustment, emphasizing the role and reaction relationship between the subject of action and the influencing factors, which provides a theoretical perspective for an in-depth discussion of the interactive process of the interaction between Huangmei Opera and placeness.

## Methodology and sample description

### Study design

This study takes Pride's structuration theory as the basic analytical framework and focuses on the dialectical relationship between structure and action in the process of constructing placeness in traditional opera [31]. The value of this

theoretical perspective lies in the following: firstly, it breaks through the limitations of the dichotomy of "structure" and 'action' in traditional research and provides a dynamic analytical tool for understanding the inheritance and innovation of Huangmei Opera, a living cultural heritage; secondly, the theory emphasizes the importance of the spatial and temporal dimensions, and can effectively capture the evolutionary trajectory of Huangmei Opera during different periods of history and in different spaces of performance; Finally, the concept of "duality" is suitable for analyzing the characteristics of traditional opera, which is both bound by traditional norms and constantly breaking through and innovating.

At the methodological level, the first is the organic combination of macro-institutional analysis and micro-practice observation, which can grasp not only structural adjustments, such as the restructuring of Huangmei Opera troupes, the promotion of globalization, and the context of modernity, but also penetrate specific scenarios, such as the practice of exhibition spaces. Secondly, it establishes a dual perspective of historical compilation and comparison of different actors, both tracing the institutionalization of Huangmei Opera from the grassroots to the theatre and comparing the differences in the practices of different types of actors; thirdly, it develops a three-dimensional analytical model of "structure-action-place," which enriches the spatial dimension of the study of traditional opera by examining the interaction of the material characteristics of the exhibition space and its symbolic meaning. This comprehensive research path offers new analytical insights into understanding the adaptive changes of traditional opera culture in modern society (Fig 1).

## Data sources

This study collected data through in-depth interviews, participant observation, and text analysis, focusing on the multidimensional dynamics of the regional construction of Huangmei Opera. From May to August 2024, the research team conducted semi-structured interviews with 35 participants in the Anqing region. Table 1 shows the demographic information

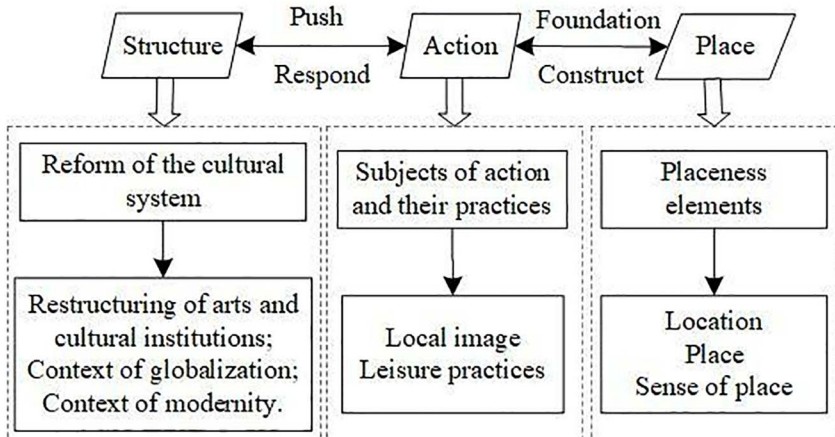

**Fig 1. The analysis framework among structure, action, and place during Huangmei Opera participating in placeness construction.**

**Table 1. Composition of the interview sample.**

| Type of respondent | Numbers | Code range | Interview highlights |
| --- | --- | --- | --- |
| Local government | 2 | ZF01-ZF02 | Policies and incentives for the protection of intangible heritage |
| Huangmei opera practitioner | 6 | CY01-CY06 | Script creative boundaries and survival strategies |
| Local resident | 17 | JM01-JM17 | Practical strategies and cultural memory |
| Cultural consumers | 10 | WD01-WD10 | Symbolic cognition and consumption behavior of Huangmei Opera |

of the participants, with a focus on the logic of policy implementation, strategies for artistic practice, and differences in cultural identity. Concurrently, participatory observation methods were employed to gather information on the cultural landscape of Huangmei Opera, its various performance forms, and related activities. Textual data included the Anqing Region Journal, local government cultural policy documents, and interviews with Huangmei Opera artists. These data were integrated to interpret the structural adjustments in actors' representations of their participation in the regional construction of Huangmei Opera and its metaphorical socio-cultural significance. All data were anonymized and coded, with sensitive information de-identified.

## Data analysis

This study adopted inductive coding for data analysis, avoiding theoretical presuppositions throughout. First, we repeatedly read the 35 transcribed interviews, field observation notes, and policy texts to establish an overall understanding of the development context of Anqing Huangmei Opera; then, we manually coded line by line to generate the initial codes and clustered the semantically related codes to form the provisional themes through the continuous comparison method. Theme validation involves a triple cross: textual, behavioral, and mutual verification is realized against interview claims and observation records, multi-perspective triangulation is achieved by comparing contradictory representations from various groups, and five key interviewees are invited to verify the accuracy of these representations. In the theme refinement stage, we iteratively optimize the naming of core themes based on the internal logic of the data and ultimately embed the mature themes into the analytical narrative of Huangmei Opera's placeness construction, where each argument is directly anchored to the original interview quotes, forming a substantive interpretation that emerges from the bottom up.

## Ethical considerations

The study was reviewed by the Research Ethics Committee of Anqing Normal University, and all procedures were approved prior to data collection. The principle of informed consent was strictly adhered to throughout the research, and participants were informed of the study's purpose, the intended use of their data, and the anonymization measures that would be implemented. Alphanumeric codes were used to replace personal information, and sensitive expressions were semantically desensitized. The raw text was stored on a password-encrypted device accessible only to the research team. All respondents were adults over the age of 18 and were explicitly informed that they could withdraw from the study at any time without conditions. The study eliminated any form of incentive to participate and strictly adhered to the Declaration of Helsinki and institutional ethical standards.

## Results

Based on the analytical framework of "structure-action-place" interaction, this study systematically examines the internal process by which Anqing Huangmei Opera has deeply integrated into and shaped placeness. It is found that the impetus of Anqing Huangmei Opera's participation in the construction of placeness stems from the in-depth interaction between structural adjustment, the practice of multiple subjects, and the elements of placeness. Within the structural framework of cultural system reform, globalization, and modernity, local government, practitioners, residents, and consumers are the primary actors who engage in dynamic practices and collaborate on utilizing Anqing's unique local resources. This process of inter-construction is manifested in three primary mechanisms: the creative vitality stimulated by the restructuring of the theaters, the shaping and dissemination of the local image of Huangmei Opera as a symbol, and the creative negotiation between traditional and modern elements.

## Structural elements: Cultural system reform

At the beginning of the founding of the People's Republic of China, the reform of traditional opera was put on the agenda. The key event in this phase was the issuance of the Directive on the Reform of Opera by the State Council of the Central

People's Government in 1951, which affirmed the educational significance of opera and put forward three significant aspects of the reform of opera: reforming the people, the opera and the system, to realize the innovation of the art. The socialist construction period in 1956 promoted the modernization of opera and encouraged the creation and performance of modern themes. Since 1978, China has initiated a cultural system reform to promote the prosperity and development of culture and stimulate cultural vitality. 1985, the General Office of the Communist Party of China (CPC) Central Committee and the General Office of the State Council forwarded the Ministry of Culture's "Opinions on the Reform of Performing Arts Organizations," which called for streamlining, merging, or abolishing performing arts organizations to carry out internal system reforms. In 1989, the Central Committee of the Communist Party of China issued "Several Opinions on the Further Prosperity of Literature and Art," proposing the retention of a small number of nationally owned troupes and the promotion of the restructuring of art organizations across various forms of ownership.

Since the new century began, rapid social and economic development has consistently elevated the importance of culture, and the reform of the cultural system has been further deepened. In 2000, the "Proposal of the Central Committee of the Communist Party of China on the Formulation of the Tenth Five-Year Plan for National Economic and Social Development" adopted by the Fifth Plenary Session of the Fifteenth Central Committee of the CPC put forward "cultural industry" as an official concept, indicating the path and direction of the reform. In 2002, the 16th CPC National Congress made new arrangements for the strategic adjustment of cultural construction and cultural industries, and the joint development of cultural undertakings and cultural industries became the focus of future work. The Provisions on the Conversion of Operational Cultural Institutions to Enterprises in the Reform of the Cultural System, released in 2018, clarified that the supportive policies already in place during the cultural system reform would be continued, such as the exemption from corporate income tax for five years after restructuring.

In the more than 40 years since the reform of the cultural system was promoted, the overall direction of work has been shaped by the insistence on promoting the differentiated development of cultural undertakings for the public good and cultural enterprises for business purposes, the protection and inheritance of traditional culture in the context of globalization, and the creative transformation and innovative development of traditional culture. Firstly, the reform of the cultural system has led to the integration and optimization of traditional theatrical performing groups' resources, the effective construction of a public cultural service system, and the closer integration of art groups with the market. Secondly, traditional culture in the context of globalization has become a representation of place brand symbols, which helps to establish and enhance cultural self-confidence, generates market demand for traditional culture, and the advantages of traditional place culture are becoming increasingly prominent. Finally, traditional culture in the context of modernity needs to undergo creative transformation and innovative development, exploring and seeking sustainable paths of inheritance for traditional culture under the guidance of people-centered policies. Huangmei Opera is a typical representative of Anqing's excellent place traditional culture. In the stage when the theater market is in the doldrums, the cultural system reform vigorously pushed forward by the state provides new opportunities for traditional opera, including Huangmei Opera, which constitutes an important external condition for Huangmei Opera to participate in the construction of placeness.

## Actor practice: Subjects of action and their practical responses

Under the impetus of cultural system reform, subjects of action respond to adjustments in structural elements through strategy formulation and daily leisure practices. In this process, the action subjects involved include local governments, Huangmei Opera practitioners, residents, and cultural consumers, and the practice behaviors of different types of action subjects differ significantly.

Local governments have strengthened place cultural brands through institutional intervention, establishing Anqing's cultural status as a core area for Huangmei opera through the construction of physical spaces such as the China Huangmei Opera Museum, Yan Fengying's former residence, and the Shipai Opera Characteristic Town, making them effective channels for residents and cultural consumers to learn about the opera. At the same time, it has taken the lead

in constructing diversified performance forms, maintaining localized inheritance in theaters, halls, and public spaces, implementing "Opera on Campus" to cultivate young audiences and reserve talents, and organizing "Opera in the Community" to promote the improvement of public welfare cultural undertakings (ZF-02), as well as promoting "Huangmei Opera+Tourism" to realize two-way development of the industry (ZF-01). Relying on the Anqing Huangmei Opera (Local Opera) Research Institute to carry out protection and creation and importing professional talents for the development of Huangmei Opera through the Huangmei Opera Art College of Anqing Normal University and the Anhui Huangmei Opera Art Vocational College.

Huangmei Opera practitioners convey the qualities of a place through their creations and performances, and professional troupes incorporate elements of place into their scripts, highlighting the optimistic spirit of a place through the telling of local stories in Anqing. Based on their traditional repertoire, private troupes, and folk groups participate in the creation of new repertoire, utilizing local history and culture as sources of inspiration. During the creative process, playwrights often emphasize the importance of seeking storylines from historical records [36] or obtaining character materials from real life through fieldwork [10]. The director integrates living and programmatic techniques in the stage design [28]. At the same time, the actors focus on understanding the essence of the characters (CY-02), using dialect recitation and living performances to present the "unique singing and living local characteristics of Huangmei Opera" (CY-05) and maintaining the "easy-to-understand, simple and intimate artistic characteristics" (CY-06), and to realize the revitalization and transmission of placeness elements in incarnational practice.

Residents strengthen their cultural identity through daily practice, and those who love Huangmei Opera regard it as a form of leisure and entertainment, expressing the emotional experience that "listening to it is very comfortable and puts people in a better mood" (JM-03). Non-active participants are still profoundly influenced by the cultural atmosphere of the opera, generally agreeing that Huangmei Opera is an important part of the city's culture and actively transforming it into a medium of social interaction; for example, "When friends from abroad come over, they will usually take them to the Guild Hall to listen to the opera, which is, after all, the biggest feature of Anqing, and is a window to show the place image of Anqing (JM-09) ". Based on the perception and identification of residents, Huangmei Opera is embedded with and transformed by placeness elements, which continuously promote the connection between Huangmei Opera and Anqing, further confirming the importance of the "local voice" [37,38].

Cultural consumers build cross-regional connections through diversified participation, "Cannot miss the big opera performances during art festivals and exhibition weeks (WD-09)", "When I take the opportunity of a business trip, I go to the guild hall to listen to the opera (WD-08)", and "Discover Huangmei Opera through the Shake Music platform (WD-10)," showing how cultural consumers connect with Huangmei Opera. Through the visual and auditory experience of Huangmei Opera, cultural consumers perceive and identify with the placeness embedded in Huangmei Opera, which is an "other" placeness constructed through the process of perceiving "Huangmei Opera" as "placeness." The audience of Huangmei Opera is getting wider and wider; not only do you Anhui people like it, but we foreigners also like it; the singing voice is sweet, the storyline is close to life, and these place characteristics must be maintained (WD-02) ", which also conveys the idea of "letting yourself know more about the traditional culture and the characteristics of each place and discovering the beauty of each place, and let your memories of each place be good (WD-07)", which effectively strengthens cultural self-confidence and highlights the core value of Huangmei Opera as a carrier of people-place relations.

## Elements of placeness: Foundations and constructs

The inheritance and development of Huangmei Opera is deeply rooted in the local environment of Anqing. Under the action of multiple forces, the placeness elements of Anqing provide the material, social, and spiritual foundations for Huangmei Opera, which embodies place characteristics and, in turn, contributes to the construction of placeness. Anqing's unique geographic environment, historical tradition, and humanistic spirit form the core foundation of Huangmei Opera's cultural production, continuing to contribute to the construction of placeness through the creation and dissemination of the opera.

The geographical location and natural environment of the Anqing area lay the material foundation for the construction of placeness. Anqing is situated at the junction of Anhui, Hubei, and Jiangxi provinces, on the north bank of the lower reaches of the Yangtze River, and at the southern foot of the Dabie Mountain, which facilitates the spread of Huangmei Opera across the region. The balanced distribution of mountains, hills, rivers, and lakes, as well as the mild climate with four distinct seasons, not only produces rich landscape resources but also provides conditions for agricultural production, forming a self-sufficient living environment.

Historical culture and folk traditions provide a source of content for the placeness of Huangmei Opera. Folk narratives, such as the Legend of the Southeast Flight of the Peacock and the allusion to Six Feet Lane, have become themes in classic plays. The spiritual qualities of historical figures such as Zhang Ying, Deng Shiru, Cheng Changgeng, and Deng Jiaxian are incorporated into characterization. Daily life practices such as production, festivals, and rituals, as well as dialect expressions, infuse the performance program with vivid place characteristics.

The humanistic spirit shapes the artistic character of Huangmei Opera, and the open and tolerant, soft, and pragmatic regional character formed by the intermingling of Wu and Chu cultures is directly reflected in the narrative logic and emotional expression of Huangmei Opera. The local sentiment cultivated by the living environment of green mountains and green water is transformed into artistic language through works on life themes, such as "Hog Grass." The people's sense of creativity in their work promotes the innovation of traditional programs in modern productions such as Sister-in-law of the Duck.

In the context of the intersection of globalization and localization, the elements of Anqing's placeness have been integrated into the cultural production of Huangmei Opera, making it a symbolic place of traditional culture that continues to participate in the process of constructing and reproducing placeness. Thus, the elements of Anqing's placeness are the roots of Huangmei Opera's cultural production, and Huangmei Opera, which incorporates these elements, in turn, profoundly influences and shapes the formation and evolution of the city's placeness.

## Role mechanisms of Huangmei Opera's participation in the construction of placeness

The interactive process of Huangmei Opera and the construction of placeness is essentially a manifestation of the dynamic interplay between structure, action, and place. The adjustment of structural elements prompts different action subjects to adopt differentiated response strategies, jointly shaping the placeness.

**Stimulation of the vitality of theatrical creativity under the restructuring of arts and cultural institutions.** The restructuring of Huangmei Opera art troupes is a critical structural adjustment influencing the development of Huangmei Opera. Local governments have played a crucial role as both responders and promoters in this process. As shown in the S1 Table, under the guidance of the central government's cultural system reform policies, the Anqing Municipal Government led the systematic restructuring of state-owned Huangmei Opera art troupes from 2004 to 2012. This included the integration and restructuring of municipal-level troupes, the establishment of the Anqing Huangmei Opera Theatre No. 1 and the Zaifen Huangmei Art Theatre in 2005, promoting the restructuring of county-level troupes and exploring models for deeper reforms, as well as the shareholding system reform pioneered by the Zaifen Huangmei Art Theatre in 2012. By the completion of the restructuring, the Yuexi County Huangmei Opera Troupe was merged into the Yuexi Gaoqiang Inheritance Center. Meanwhile, the remaining eight troupes also completed their restructuring. After restructuring, the troupes operated in the market with policy support, significantly enhancing their creative vitality in repertoire development; they actively implemented the action strategy of "singing upward, going downward, and going outward" to respond to reform requirements.

At the same time, the market vitality of private troupes has been effectively released. According to the data from the Anhui platform of the National Enterprise Credit Information Publicity System, as of January 2024, the number of registered private troupes of Huangmei Opera in Anqing has reached 116. These troupes perform flexibly, with repertoire content close to real life, and are active in parks, squares, streets, alleys, and rural areas, as well as in the markets of

neighboring provinces and cities, with qualities that are highly compatible with the fresh, rustic, and natural style inherent in Huangmei Opera. In response to the government's policy of encouraging original repertoire, some private troupes have begun to shift from singing traditional classics to creating their repertoire. For example, the play "Hu Jiugen Leaving Office," created by Wangjiang Changjiang Huangmei Opera Communication Co., Ltd., is based on the real environment of Shitang Village in Wangjiang County and takes Hu Jiugen, the local village branch secretary, as the prototype of the character, which shows the noble quality of grass-roots cadres' courageous introspection and dedication to serving the people and embodies the pure and simple creative characteristic of its rootedness in life.

The restructuring of Huangmei Opera troupes has led to the exploration of market-oriented development paths, actively seeking and stimulating audience demand. Practitioners have vigorously launched newly created productions based on rearranging traditional repertoire, effectively changing the audience's singular impression of Huangmei Opera. As the hometown of Huangmei Opera, Anqing's intense opera atmosphere not only provides residents with a venue for leisurely practice but also serves as an important space for residents to engage in social interactions. Residents expressed their strong identification with the local characteristics of Huangmei Opera through their participation, with some respondents stating that "everyone gets together to entertain themselves, whether traditional or modern, it is all Huangmei Opera, and her distinctive chants and singing tunes are there (JM-14)." Some interviewees also mentioned that "when a friend comes to visit, the first thing that comes to mind is to take him to an authentic Huangmei opera; after all, it is his hometown opera (JM-04)". For foreign cultural consumers, real-life Huangmei opera plays that fit modern aesthetics and are close to their daily lives have become a medium for them to establish an emotional connection, with one foreign consumer stating, "Modern Huangmei opera is closer to our lives and gives us positive energy, telling us that no matter what kind of suffering we have to go through, we need to rise to the occasion, and to keep our beginner's heart and good heart (WD -03)." It is noteworthy that more cultural consumers from the younger generation are involved in the consumption circle of Huangmei Opera, which is no longer exclusive to the elderly but is gradually becoming a new fashion of modern consumption under the joint action of multiple stakeholders. The perception of residents and cultural consumers of the placeness elements contained in Huangmei Opera, in turn, further promotes the enhancement of the vigor of practitioners' theatrical creations.

**Shaping the image of places in the context of globalization.** As a representative of Anqing's deep cultural heritage, Huangmei opera has become a core cultural text for local governments to construct the city's unique placeness. The local government has taken the lead in creating a cultural landscape for Huangmei Opera, which is not only a key initiative to protect and preserve the local traditional culture but also the foundation for fostering an intense atmosphere for local opera. Local governments not only provide policy support for professional troupes' theater performances but also create convenient conditions for private troupes to use public spaces such as park squares for performances. At the same time, they actively promote the integration of Huangmei Opera into the education system, not only cultivating professionals in colleges and universities but also reaching out to primary and secondary schools through the opera on campus campaign, which has significantly increased residents' response to the government's cultural practice initiatives.

Huangmei opera practitioners, including playwrights and performers, are the core body of play creation and rehearsal. In the process of the local government's active branding of Anqing as a hometown of opera, they responded to the government's marketing strategy at the level of text production. First, in order to strengthen the connection between Huangmei Opera and the local community, practitioners "focus on in-depth excavation of local history, culture, and real-life experiences (CY06)" and "organize fieldwork for playwrights and performers (CY02)" to collect rich local materials and achieve the goal of singing local stories with local plays. This is reflected in the reproduction of local historical figures in plays such as "Southeast Flight of the Peacock," "Big Qiao and Little Qiao," "The Virtuous Prime Minister of the Qing Dynasty," "Xu Xilin," and the presentation of local real life in plays such as "Woman under Duxiu Mountain," "Hu Jiugen Leaving Office," and "Duck's Sister-in-law." Through the creative practice of the practitioners, elements of placeness, such as geographical location and places in Anqing, are transformed into physical environments and social situations on stage to complete the artistic expression of placeness. Secondly, Huangmei Opera creates plays with the main tone of promoting

 

positive energy, and the emotions and meanings are conveyed through the exquisite story design and physical performances of the actors. For example, "Deng Jiaxian" from the Zaifen Huangmei Art Theater sings about hometown heroes with hometown operas, celebrating the country's feelings. The Anqing Huangmei Opera Art Theater's "Boat Offering" is based on a real-life incident in which Zhanghu Hui villagers offered their boats to help the army during the Battle of the River Crossing, reflecting the righteousness of ordinary people. Whether portraying historical figures or depicting folk life, these works convey positive energy and follow the tradition of Huangmei Opera, which views the big picture from a small perspective, often beginning with love and affection and ultimately transcending to hometown and national sentiment.

In the context of globalization, local governments and practitioners have actively explored placeness elements to integrate into the cultural production of Huangmei Opera. However, their effectiveness needs to be tested by the interpretation of residents and cultural consumers. On the one hand, residents who actively participate in Huangmei Opera performances become part of the cultural landscape while watching the performances, and the process of perceiving placeness is the process of constructing placeness. Residents who do not actively participate in the performances also live in the rich cultural atmosphere of Huangmei Opera created by the local government, practitioners, and enthusiasts, and practice the brand identity of the hometown of opera through the practical action of active learning. For example, interviewees JM07 and JM15 both mentioned that they "let their children learn Huangmei Opera as a specialty" and believe that this specialty is unique.

On the other hand, cultural consumers experience Huangmei Opera, perceive the local qualities embedded in it, and see it as a way to relieve stress and relax. One consumer, WD01, said, "Many of the Huangmei opera performers are from Anqing, with sweet singing voices and handsome costumes that are pleasing to the eye, and listening to Huangmei opera can be a moment of relaxation and letting go when you are under stress." This suggests that Huangmei Opera has become an important medium for consumers to experience and perceive the placeness of Anqing.

**Negotiating tradition and modernity in the context of modernity.** In the context of modernity, members of society negotiate between tradition and modernity through active social and cultural practices in order to find a balance [39]. This process is notable in the development of Huangmei opera.

Local governments actively encourage professional troupes to distill themes from traditional culture and contemporary life and to create and stage new Huangmei opera plays. Under the guidance of the policy, professional troupes have created themes that closely match the themes of the times, such as "Duck's Sister-in-law," which reflects poverty alleviation, "Deng Jiaxian," which glorifies science in the service of the nation, and "Immortal Poplar," which showcases the history of the revolution. The integration of placeness and era has become a core theme that practitioners actively explore, with local locations and venues forming the basis for creation, thus realizing the sublimation of emotions and values. At the same time, the local government emphasizes the role of interaction within and outside the province and the linkage of cities and counties, encourages troupes to go out on exchange tours, and invites troupes from outside the province to participate in local festivals and events, aiming to break through regional limitations, establish a sense of "Greater Huangmei", and effectively promote the strategy of Huangmei Opera going out.

Based on this, Huangmei opera creators and performers began to explore foreign markets to expand their audience base, integrating local cultural characteristics and adapting to the preferences and needs of different audience groups. Fashionable elements increasingly appear in the repertoire. For example, the elegant Huangmei Opera "Six Memoirs of a Floating Life" incorporates modern elements while retaining the essence of classical opera, thereby breaking through the previous vulgar and straightforward style of Huangmei Opera and presenting a new face of elegance and popularity. Famous Huangmei Opera artist Han Zaifen has repeatedly emphasized the need for Huangmei Opera to "keep up with the times" and "be fashionable." This reflects the fact that under the new situation of deepening the creative transformation and innovative development of traditional culture, local governments have provided practitioners with freer creative space and encouraged the inheritance and innovation of repertoire. Market-oriented demand has become the creative guide, and the consciousness of "Greater Huangmei" has gradually formed. The traditional essence and fashionable elements are intertwined and symbiotic in the repertoire, jointly constructing a shared sense of place.

However, this innovation has provoked mixed audience reactions. Residents and consumers who agree believe that realistic themes fit the essence of opera's closeness to life, while critics argue that it has lost the true nature of traditional art. Amid the heated debate over-praise and criticism, China Central Television recognized the contributions of newly created plays to the innovative development of traditional culture. Faced with the support and encouragement of structural adjustment, local governments and practitioners have actively explored innovative paths, on the one hand, while paying attention to the core values sought by residents and consumers, on the other, striving to achieve an effective balance between the tradition and modernity of Huangmei Opera. With the development of the times, residents and consumers have come to realize that Huangmei Opera, which clings to the past without looking to the future, will eventually decline, so they have also expressed their expectations for newly created themes, emphasizing that only by constantly pushing the boundaries of what is new can they always have something to watch. Interviewee JM11 pointed out that "Huangmei Opera itself is very inclusive; she can be vulgar and elegant." At the same time, innovation can not forget the fundamentals; for the audience, singing, characters, emotions, and spirit are the key elements in the formation of placeness perception. On this basis, the expansion of the subject matter and the exploration of the play align with the development orientation of righteousness and innovation, reflecting the open-mindedness of the main body of action in the protection and inheritance of Huangmei Opera and the pursuit of progress in the field.

## Discussion

Based on the perspective of structuralization theory, the study reveals the interweaving of multiple dynamics and the dynamic, practical nature of Huangmei Opera's participation in the construction of placeness. The study breaks through the dualistic framework of "structure-action" and focuses on the complex process of continuous mutual construction between Huangmei Opera and placeness. In the context of modernity, this construction displays the generative characteristics of "sharing" and "progress" of placeness, reflecting the dynamic negotiation between tradition and modernity. In the face of the challenges posed by globalization, the path of Huangmei Opera's living inheritance demonstrates a creative balance between its local roots and innovative transformation. Huangmei Opera is not only the bearer of a placeness structure but also the key medium for recreating placeness structure through the practice of multiple subjects continuously.

### Intertwined dynamics: Breaking through the "structure-action" dichotomy

This study transcends the dichotomy of structure and action in traditional studies of placeness, revealing the complex association and synergy between the dynamic elements of Huangmei Opera's participation in the construction of placeness from the perspective of structuration theory. Huangmei Opera is not a static text passively reflecting placeness but is in a continuous process of mutual construction with placeness. As pointed out by related studies, there is a dynamic and complex interplay between festivals, local industries, and placeness [40]. This study focuses on the specific cultural text of Huangmei Opera, which further confirms the universality of this inter-constructive mechanism. The repertoire content, performance form, emotional expression, and the regional consciousness of "Greater Huangmei" carried by Huangmei Opera are not only the products of local cultural traditions but also continue to shape, strengthen, and even reconstruct Anqing's place identity and cultural space through the guidance of the government, the interpretation of the actors, and the practice of the audience. This dynamic inter-constructive relationship not only expands the connotation and extension of the "text" in the study of placeness construction but also provides key theoretical support for understanding the sustainable development of Huangmei Opera and its central role in the construction of local society and culture. Structuration theory emphasizes "structural duality"; that is, social structure is both the mediator and the result of social practice. Huangmei Opera is a vivid embodiment of this structural duality, as it is both the bearer of placeness structures and the key medium of action for reproducing or transforming them through the practices of the actors.

## Consultations on modernity: Placeness production of "sharing" and "progress"

Within the context of modernity, Huangmei Opera's participation in the construction of placeness exhibits distinctive dynamic characteristics centrally embodied in "shared placeness" and "progressive placeness." This does not mean a simple abandonment of tradition but a contemporary response to the regional consciousness of "Greater Huangmei," a creative fusion of traditional essence and modern elements in practice. The concept of placeness itself is not rigid and unchanging. Instead, it continues to evolve, presenting a complex situation in which tradition and modernity are intertwined and negotiated with one another. This strongly confirms the academic view that modernity exhibition and placeness expression can be embedded and transformed into each other [5]. From the viewpoint of structuration theory, the subjects of action continue to engage in reflective monitoring as they practice inheriting and developing Huangmei Opera. They selectively absorb placeness elements based on their understanding of the current social context and their expectation of future development: on the one hand, they prudently identify and maintain those core elements that are regarded as cultural roots and need to be kept relatively stable; on the other hand, they actively embrace change, exploring those elements that can be adapted to modern aesthetics, integrated into the spirit of the times, and realized in innovative transformation. This continuous process of selection, adaptation, and innovation is the core driving force behind the dynamic construction of the relationship between Huangmei Opera and placeness, highlighting the mobility of actors under structural constraints.

## Path of living heritage: Balancing place roots and global perspective

In the face of the challenge of cultural homogenization brought about by globalization, the living inheritance strategy of Huangmei Opera profoundly embodies the logic of structuration theory on the use of rules and resources. The first strategy is to plant placental roots. The survival and development of Huangmei Opera cannot be separated from the nourishment of its birthplace and core transmission area. By establishing a close "audience connection" with the local society, Huangmei Opera not only fully absorbs local cultural nutrients but also profoundly influences the creation of local theater cultural ecology, the generation of cultural experience atmosphere, and the shaping of local cultural brand image through continuous performance practice, and this two-way interaction is the basis of its practice cycle. Secondly, it is crucial to insist on the continuity and evolution of cultural expression. Huangmei Opera is by no means a frozen specimen of time and space, and its "living" nature lies in the fact that it is a dynamic cultural practice process of continuous change. The creation of an elegant artistic atmosphere enhances its aesthetic qualities, the content of the repertoire closely follows the pulse of the times and highlights its vitality, and the expression of emotions, from "small love" to "great love," reflects the enhancement of its spiritual realm. The evolution of this cultural expression is the core connotation of living heritage. Ultimately, realizing a creative balance between tradition and modernity is the key path to the sustainable development of Huangmei Opera in modern society. This is manifested in the following ways: actively using modern stage technology to enhance artistic expression, continuously expanding the range of topics to respond to the issues of the times, flexibly handling language expression to adapt to a broader audience, and personalized interpretation of characters by actors based on their personal experience and knowledge, and injecting emotion into their creative interpretation. This kind of negotiation is not a departure from placeness but rather a creative transformation and innovative development of Huangmei Opera's localized inheritance under the premise of rooting in local cultural genes and through the continuous updating of technology, materials, knowledge, and experience to ensure that it maintains its vitality and vigor in a changing world. This is precisely the practical process by which actors utilize the resources and rules within the established structure and create new ones.

## Conclusions

The complex and dynamic interaction between traditional opera and placeness has increasingly become an important topic for academic discussion. Taking Anqing Huangmei Opera as a specific case, this study, based on the theoretical

perspective of the mutual construction of text and placeness, innovatively introduces and applies the interactive analytical framework of "structure-action-place" and reveals the intrinsic mechanism of Huangmei Opera's participation in the construction of placeness in depth. The study finds that the process of Huangmei Opera's in-depth integration into and shaping of placeness is essentially the inevitable result of continuous interaction and mutual construction through structural adjustments, the practice of multiple action subjects, and the specific elements of placeness.

In this dynamic mechanism, structural adjustments at the macro level constitute the key initial conditions and regulatory framework. This is mainly reflected in the reshaping of cultural institutions by the state-driven cultural system reform, as well as the profound impact of the wave of globalization and the context of modernity. As a key player, the local government has actively responded to and guided these structural changes, encouraging the Huangmei Opera Theatre to complete the restructuring and optimization of its internal system through policy support and resource allocation. In this context, Huangmei Opera has been consciously built as the core cultural brand symbol of the city of Anqing. This strategic positioning has prompted the Huangmei Opera not only to carry and express the deep placeness qualities and highlight Anqing's unique historical traditions, dialectal charms, and humanistic spirit but also to take the initiative to incorporate elements of modernity and explore innovations in stage presentations, narrative styles, and emotional expressions in order to adapt to the needs of the wider modernized cultural consumption market.

At the same time, the dynamic practice of multiple action subjects is the core engine driving the mechanism's operation. Huangmei Opera practitioners, including playwrights, directors, actors, etc., have been significantly stimulated. They actively respond to structural changes and market demand, devote themselves to the innovative creation and arrangement of repertoire, and strive to find a balance between artistic heritage and innovation. Local governments provide institutional guarantee and development direction for the reform and artistic innovation of theaters through planning guidance and policy support. Residents' perception and recognition of Huangmei Opera in their daily lives is the social foundation for the internalization and maintenance of placeness. The cultural consumer's choice of performance and market feedback directly constitutes the market dynamics and value judgment of Huangmei Opera development, constantly shaping the creative direction and expression of the repertoire. These different levels and types of actors, with their logic of practice, jointly weave the practical network of Huangmei Opera's participation in the construction of placeness.

Specific placeness elements constitute the material foundation and cultural background of this mutual construction process. Anqing's unique geographical environment, historical deposits, dialect system, folk customs, and collective memory provide an inexhaustible source of creation and spiritual core for Huangmei Opera. In turn, through its continuous performance activities and cultural dissemination, Huangmei Opera constantly selects, refines, strengthens, and re-interprets these placeness elements, transforming them into perceptible and experienceable cultural symbols, thus profoundly participating in and promoting the construction of place cultural identity, the shaping of place image and the creation of place cultural atmosphere. Placeness elements are not only the soil on which Huangmei Opera survives but also the core content of its artistic expression, and at the same time, they are constantly redefined and given new meanings in the performance and dissemination of Huangmei Opera.

Therefore, this study ultimately reveals that Huangmei Opera's participation in the construction of placeness is not a unidirectional reflection or simple attachment of cultural labels but rather a complex socio-cultural process in which structure, action, and place are deeply intertwined. This process deeply embodies the core idea of Pride's structuration theory: macro-structures provide rules and resources for action, shaping the boundaries of possibilities for practice; multiple actors, by their mobility, interpret, apply, and even change the structures in practice and continuously construct placeness, and placeness itself is not only the specific spatial and temporal context in which the practice of this inter-construction occurs but also the object and product of its continuous action. The case of Anqing Huangmei Opera not only vividly explains the inner mechanism of traditional opera's living inheritance during the process of modernization but also provides important theoretical inspiration and practical reference for understanding the resilience, adaptability, and creative development of place culture in the era of globalization (Fig 2).

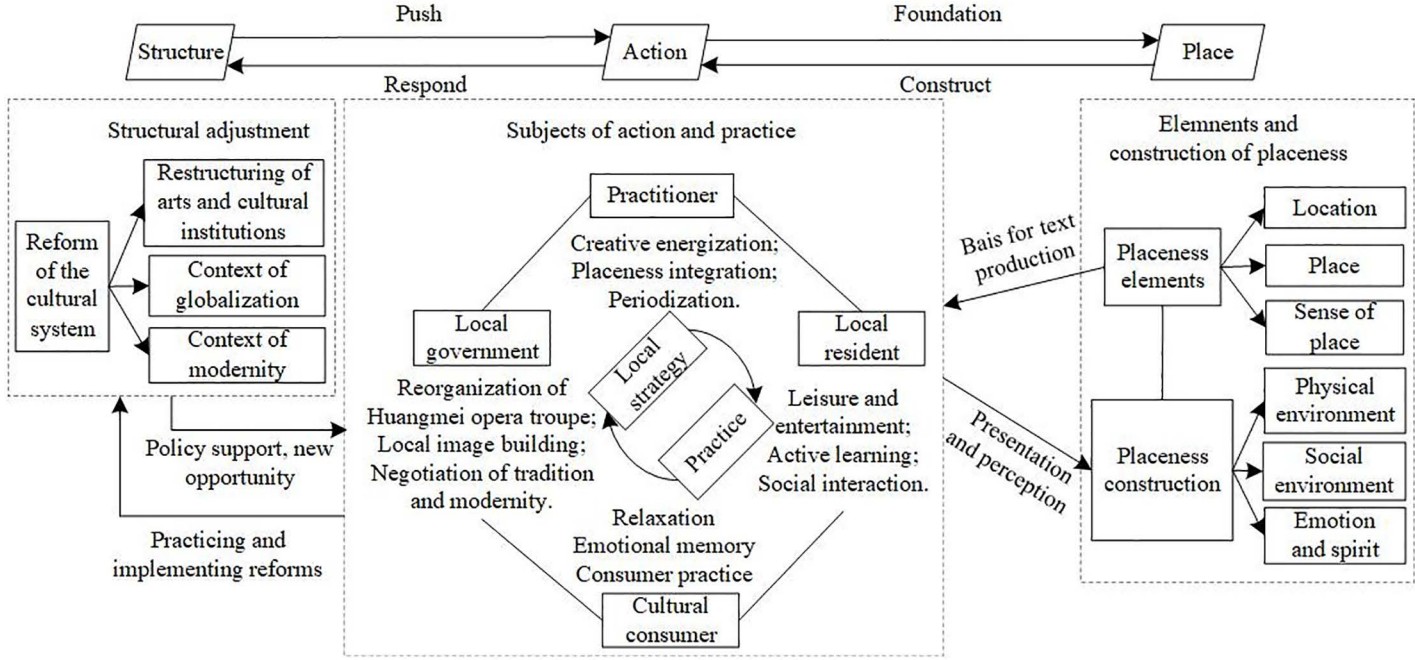

**Fig 2. The mechanism of Huangmei Opera participating in placeness construction under the interaction of structure, action, and place.**

## Limitations of the study

The current study has several limitations. This study is an exploration based on the "relationship between text and placeness," and subsequent studies can focus on the perspective of "place" that depends on Huangmei Opera and explore the mechanism of constructing placeness based on the theoretical foundation of "space and place." At the same time, the subjects of action included in this study can be further refined to explore in depth the practices and pathways of the various subjects of action involved in the construction of Huangmei Opera's placeness. In addition, the development history and evolution patterns of different traditional operas vary, and the resulting relationship with "place" is characterized by individuality and complexity. Therefore, whether conclusions drawn from a single case can be extended to other traditional operas needs to be empirically tested.

## Supporting information

**S1 File. Interview outline.**
(DOCX)

**S1 Table. General situation of professional troupe in Anqing.**
(DOCX)

## Acknowledgments

We want to thank the staff at Anqing Normal University for their assistance with data collection and analysis. We would also like to thank all the women and men who agreed to participate in this study, as well as the staff who assisted us in recruiting participants.

## Author contributions

**Conceptualization:** Fang ZHU.

**Data curation:** Fang ZHU, Qin Su, Jiahui Xu, Lei Zhu.

**Formal analysis:** Fang ZHU, Qin Su.

**Funding acquisition:** Fang ZHU, Lei Zhu.

**Investigation:** Fang ZHU, Jiahui Xu.

**Methodology:** Jiahui Xu, Lei Zhu.

**Project administration:** Fang ZHU, Jiahui Xu, Lei Zhu.

**Visualization:** Lei Zhu.

**Writing – original draft:** Fang ZHU.

**Writing – review & editing:** Fang ZHU, Qin Su, Lei Zhu.

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
