## [Decision Letter · Decision Letter 0]

8 May 2025

PONE-D-25-06193The Construction of Placeness in Traditional Opera from the Perspective of Structuration Theory: A Case Study of Huangmei Opera in Anqing, ChinaPLOS ONE

Dear Dr. ZHU,

Thank you for submitting your manuscript to PLOS ONE. After careful consideration, we feel that it has merit but does not fully meet PLOS ONE’s publication criteria as it currently stands. Therefore, we invite you to submit a revised version of the manuscript that addresses the points raised during the review process.

Your manuscript has been reviewed by two reviewers with the aim of providing a relatively objective assessment of the article. Both reviewers have recommended a major revision. After reviewing the specific comments, we agree with the reviewers and therefore the editorial decision is: Major revision.

**Please find the reviewers' comments at the end of this email, which explain in detail the reasons for rejecting the article. I have selected some of the key points below:**

**In general, the paper needs better organization.****In the introduction, it would strengthen the argument on the methodology if it would be clearly stated what is innovative about this specific methodological development.****First part of the Methodology section sounds like an extension of the Literature Review. Generally, sections should have their own distinct character, theme, aspect or angle of the research presentation****Results and discussion is too short. Need the give exactly discussions section.****Results section need to make separate from discussion and give whole results for discussion part.**

**It was also mentioned that authors should verify that all citations are indeed relevant.**

We look forward to receiving your revised manuscript.

Kind regards,

Václav Venkrbec, Ph.D.

Academic Editor

PLOS ONE

**Journal Requirements:**

1. When submitting your revision, we need you to address these additional requirements. Please ensure that your manuscript meets PLOS ONE's style requirements, including those for file naming. The PLOS ONE style templates can be found at https://journals.plos.org/plosone/s/file?id=wjVg/PLOSOne_formatting_sample_main_body.pdf and https://journals.plos.org/plosone/s/file?id=ba62/PLOSOne_formatting_sample_title_authors_affiliations.pdf 2. Thank you for stating in your Funding Statement: This research was supported by the Key Projects of Scientific Research in Universities of Anhui Province(2023AH050461); Major Programs of Scientific Research in Universities of Anhui Province(2024AH030073); Social Science Innovation and Development Research Project of Anhui Province(2024CX060). Please provide an amended statement that declares *all* the funding or sources of support (whether external or internal to your organization) received during this study, as detailed online in our guide for authors at http://journals.plos.org/plosone/s/submit-now.  Please also include the statement “There was no additional external funding received for this study.” in your updated Funding Statement. Please include your amended Funding Statement within your cover letter. We will change the online submission form on your behalf. 3. We note that your Data Availability Statement is currently as follows: All relevant data are within the manuscript and its Supporting Information files. Please confirm at this time whether or not your submission contains all raw data required to replicate the results of your study. Authors must share the “minimal data set” for their submission. PLOS defines the minimal data set to consist of the data required to replicate all study findings reported in the article, as well as related metadata and methods (https://journals.plos.org/plosone/s/data-availability#loc-minimal-data-set-definition). For example, authors should submit the following data: - The values behind the means, standard deviations and other measures reported;- The values used to build graphs;- The points extracted from images for analysis. Authors do not need to submit their entire data set if only a portion of the data was used in the reported study. If your submission does not contain these data, please either upload them as Supporting Information files or deposit them to a stable, public repository and provide us with the relevant URLs, DOIs, or accession numbers. For a list of recommended repositories, please see https://journals.plos.org/plosone/s/recommended-repositories. If there are ethical or legal restrictions on sharing a de-identified data set, please explain them in detail (e.g., data contain potentially sensitive information, data are owned by a third-party organization, etc.) and who has imposed them (e.g., an ethics committee). Please also provide contact information for a data access committee, ethics committee, or other institutional body to which data requests may be sent. If data are owned by a third party, please indicate how others may request data access. 4. One of the noted authors is a group or consortium. In addition to naming the author group, please list the individual authors and affiliations within this group in the acknowledgments section of your manuscript. Please also indicate clearly a lead author for this group along with a contact email address. 5. Please include your full ethics statement in the ‘Methods’ section of your manuscript file. In your statement, please include the full name of the IRB or ethics committee who approved or waived your study, as well as whether or not you obtained informed written or verbal consent. If consent was waived for your study, please include this information in your statement as well.

Reviewers' comments:

Reviewer's Responses to Questions

**Comments to the Author**

1. Is the manuscript technically sound, and do the data support the conclusions?

Reviewer #1: Yes

Reviewer #2: Partly

2. Has the statistical analysis been performed appropriately and rigorously? 

Reviewer #1: Yes

Reviewer #2: No

3. Have the authors made all data underlying the findings in their manuscript fully available?

Reviewer #1: Yes

Reviewer #2: Yes

4. Is the manuscript presented in an intelligible fashion and written in standard English?

Reviewer #1: Yes

Reviewer #2: No

5. Review Comments to the Author

**Reviewer #1: ** Please follow the journal author instructions. It would be useful for the reader to follow it. In general, the paper needs better organization.

In the introduction, it would strengthen the argument on the methodology if it would be clearly stated what is innovative about this specific methodological development.

In line with the latter, also balance and articulation of the methodological approach needs to be improved so that the application can be actually demonstrative of the validity of the methodology.

Improve the title.

Use different keywords from title. Improve the keywords.

I propose to the authors to be more specific, explanatory and simplified in order to be easily understandable from the readers.

Towards sustainable cities: Evaluating the distribution and functionality of green spaces in Atakum. Comprehensive risk assessment for identifying suitable residential zones in Manavgat, Mediterranean Region. The effects of Sinop province’s relative humidity values on bioclimatic comfort and urban and landscape planning. Integrating of settlement area in urban and forest area of Bartin with climatic condition decision for managements.

Wind Speeds of Bartın: Impacts of Climate Change and Settlements.

Determination of the impacts of urban-planning of the urban land area using GIS hotspot analysis.

Determining the suitable settlement areas in Alanya with GIS-based site selection analyses.

Results and discussion is too short. Need the give exactly discussions section.

Results section need to make separate from discussion and give whole results for discussion part.

Conclusion section is short and need to improve.

Correct references in the text and the reference list according to the journal's format.

**Reviewer #2:**  Overall, the Authors present an interesting subject that brings out important learning points on how placeness could help to shape a place’s cultural identity, significance, and influence, in the context of promoting traditional cultural heritage while embracing modernity in a globalized setting. This is a relevant topic for other places as well, because it provides critical lessons for "nondescript" places or countries to prop themselves, especially in the developing world. Therefore, it is a worthwhile manuscript for publication. However, the following aspects need to be attended to, for the manuscript to be more appealing to the reader.

1. Use of technical jargon and conveyance of clear points for the reader should be balanced. This could improve communication of the research process and findings to field experts and non-experts or mere field practitioners.

2. Use of standard English, as recommended by PLOS ONE, helps to observe word economy, clarity of the message, and text conciseness. This helps even where manuscript word limit is imposed. For example, a sentence such as, “Therefore, it is necessary to comprehensively consider the diversified subjects of the construction of placeness…”, can be rewritten as, “Therefore, it is necessary to comprehensively consider the diversified subjects of placeness construction…”. In this case, two words are eliminated but the message is conveyed in a concise manner.

3. The last paragraph of the introduction could be enhanced, by explicitly stating the knowledge gap(s) in the literature. At the same time, to arouse the reader’s interest, the aim/objectives/hypotheses of the paper should be clear. Further, he paper’s main contribution to the literature could be highlighted, as a punchline for the paper’s significance. Then the rest of the paper tries to show how you arrived at you claimed outcomes.

4. Sections and subsections not numbered. As the manuscript currently stands, it is easy for the reader to lose sense of the section they are reading, because the main threads of the arguments in the manuscript sound similar in many sections. Numbering sections and subsections could help to enhance the logical flow of thoughts.

5. First part of the Methodology section sounds like an extension of the Literature Review. Generally, sections should have their own distinct character, theme, aspect or angle of the research presentation. Even though most of the research process is well explained (e.g. data collection), a table could assist the reader to properly follow and contextualize the data sources, and link that to the kind of data they provided, whether it was for the structural, action or place aspects of the research in this case.

6. Figure locations indicated within sentences. Is it not possible to show these in separate lines bounded by parentheses or square brackets between paragraphs? For example, [Fig. 1 to be inserted here]. Is it not a requirement to state the Figure sources? i.e. whether copied or self-generated?

7. First sentences in some paragraphs of the Results section appear to be subsection headlines. Otherwise they don’t seem to convey any message and they could be eliminated since they are redundant.

8. Not sure if the Huangmei Opera location description fits well in the Results section. Could this be placed in another, or in a completely new section?

9. The important theoretical/historical backgrounds presented in the Results section should be minimized, and probably transferred to the Literature Review section.

10. For avoidance of doubt, an acronym such as CCTV should spelt out in full first time. Otherwise it could make the reader wonder whether the Authors were referring to Closed Circuit Television or China Central Television.

11. Lastly, sentence structure and grammar should be improved upon throughout the manuscript, to ensure clarity of thought and better communication.

6. PLOS authors have the option to publish the peer review history of their article (what does this mean? ). If published, this will include your full peer review and any attached files.

**Do you want your identity to be public for this peer review?** For information about this choice, including consent withdrawal, please see our Privacy Policy .

Reviewer #1: No

Reviewer #2: **Yes: ** Obed Chanda

---

## [Author Response · Author response to Decision Letter 1]

20 Jun 2025

Reviewer 1

1.Please follow the journal author instructions. It would be useful for the reader to follow it. In general, the paper needs better organization.

Response to Q1�We sincerely appreciate the valuable suggestions provided by the reviewers. We fully accept the recommendations regarding the need to better adhere to the journal's author guidelines and to enhance the paper's organization. In the revised manuscript, we have strictly followed the journal's latest author guidelines, meticulously reviewing and revising all details, including formatting, citation standards, and figure and table requirements, to ensure full compliance with the journal's requirements. Additionally, we have conducted a thorough review and optimization of the paper's overall structure, logical flow, and paragraph coherence. This includes re-examining the focus of the introduction and discussion, the clarity of the methods description, the presentation of results, and the conciseness of the conclusions. Our goal is to significantly enhance the paper's readability, coherence, and argumentative strength, thereby aligning it more closely with journal standards and making it easier for readers to understand. The revised sections can be viewed in the "Revised Manuscript with Track Changes."

2.In the introduction, it would strengthen the argument on the methodology if it would be clearly stated what is innovative about this specific methodological development.

Response to Q2: We sincerely thank the reviewers for their valuable comments. We fully agree that it is important to clarify the innovative nature of the methodology in the introduction to strengthen the methodological argument. To this end, we have made the following additions and improvements to the introduction in the revised manuscript:

In the third paragraph, we have added an introduction to the three-dimensional interactive model of structured theory that we have adopted, clearly pointing out its creative transformation as a theoretical paradigm. It not only breaks through the familiar dichotomy between structure and action in opera research but also introduces a dynamic perspective of “place as a process,” providing a new cognitive framework for understanding the modern transformation of traditional opera.

In the fifth paragraph, we explicitly state that, based on constructivist epistemology, we adopt a process-oriented qualitative analysis approach aimed at capturing the dynamic relationships among the three dimensions. We emphasize that this qualitative research design not only closely aligns with Pred's theoretical focus on "process" and "situationality" but also transcends cultural appearances to provide a profound elucidation of the complex logic underlying the construction of placeness.

In the sixth paragraph, we provide a supplementary explanation of the paper's overall structure to help readers better understand the argumentative framework and content organization of this study.

3. In line with the latter, also balance and articulation of the methodological approach needs to be improved so that the application can be actually demonstrative of the validity of the methodology.

Response to Q3: We appreciate the reviewers' important suggestions regarding the need to improve the balance and coherence of the methodology to demonstrate its validity effectively. We deeply understand the critical role that consistency and mutual support among the theoretical perspective, data manipulation, and analytical techniques play in demonstrating the rigor and validity of the research. To this end, in addition to strengthening the description of methodological innovation in the introduction, we have made the following key revisions in the "Methodology and sample description" of the revised manuscript:

(1)In the research design: The research value of Pred's structured theoretical perspective is explained in detail and closely combined with the subject and specific research content of this study, the applicability of this methodology in solving the research problems, and the design ideas of its analytical framework are systematically discussed to ensure that the theoretical framework is closely linked to the empirical analysis objectives.

(2) In the data source: The description has been enhanced to present the specific methods and sources of data collection, as well as the core content, more clearly and systematically, thereby improving the transparency and operability of the data acquisition process.

(3) Supplementary data analysis: This section provides a detailed introduction to the inductive coding process used in this study, explaining the specific steps involved in gradually forming, refining, and validating themes from the raw data, thereby ensuring the traceability of the analysis process.

(4) Supplementary ethical considerations: Clearly state that the study has been reviewed and approved by the Research Ethics Committee of Anqing Normal University and that all research procedures comply with the principle of informed consent. Participants are fully informed of the study's purpose, the use of their data, and the strict anonymization measures in place to ensure the standardization and ethical nature of the research process.

4.Improve the title.Use different keywords from title. Improve the keywords.

Response to Q4: We want to express our gratitude to the reviewers for their constructive suggestions regarding the improvement of the title and keywords. We fully recognize that a clear, accurate, and informative title structure, along with appropriate keywords, is crucial for enhancing the discoverability, readability, and overall structural clarity of the paper. In response to this feedback, we have systematically optimized the title hierarchy and keywords in the revised manuscript:

(1)Clarification of the overall structure: We reorganized and standardized the first-level headings, clearly setting them as Introduction, Literature Review, Methodology and sample description, Results, Discussion, Conclusions, and Limitations of the study. This ensures that the overall framework of the paper adheres to academic norms and is logically coherent.

(2) Refined chapter content: The literature review includes "Placeness formation mechanisms" and "Relationship between text and placeness," focusing more precisely on core theoretical issues. The methodology and sample description include "Study design," "Data sources," "Data analysis," and "Ethical considerations," significantly enhancing the transparency, clarity, and practicality of the methods section. The results include Structural elements: cultural system reform, Actor practice: Subjects of action and their practical responses, Elements of placeness: foundations and constructs, and Role mechanisms of Huangmei Opera's participation in the construction of placeness. These subheadings succinctly summarize the core findings and highlight the three-dimensional analytical framework of the structured theory. The Discussion includes Intertwined dynamics, breaking through the "structure-action" dichotomy, Consultations on modernity, placeness production of "sharing" and "progress," and the Path of living heritage, balancing place roots and global perspective. These titles not only distill the core arguments but also effectively link theoretical breakthroughs, reflections on modernity, and heritage protection pathways, demonstrating the depth and breadth of the Discussion.

(3) Keyword optimization: In conjunction with revising chapter titles, we carefully evaluated and improved keywords to ensure that they more comprehensively and accurately reflect the paper's core research subjects, theoretical perspectives, core concepts, and research contributions.

5.I propose to the authors to be more specific, explanatory and simplified in order to be easily understandable from the readers.

Response to Q5: We sincerely thank the reviewers for their valuable suggestions, which emphasize the need for greater specificity and conciseness in explanations to enhance reader comprehension. We fully agree that clear and concise expression is crucial for academic communication. To address this feedback, we have systematically reviewed the entire text, with a particular focus on revising the “Results.” We have refined core points, removed redundant information, optimized sentence structure, and improved logical coherence to ensure that the presentation of research findings is more focused, direct, and easy to understand.

6.Results and discussion is too short. Need the give exactly discussions section.

Response to Q6: We are grateful to the reviewers for pointing out that the results and discussion were insufficient in length and needed more precise discussion. We fully understand the importance of elaborating on the results and engaging in in-depth theoretical dialogue in order to demonstrate the value of the research. To this end, in the revised manuscript, we have optimized the structure and expanded the content of these two sections.

(1)Results section: We merged the two sections previously titled "Driving factors of Huangmei Opera's participation in placeness construction" and "Mechanisms of action." This adjustment was based on the fact that both sections were directly derived from data analysis and were logically closely related. The driving factors form the foundation for the mechanisms to operate, while the mechanisms are the concrete manifestation of the driving factors. The merged section presents the core findings in a more focused and coherent manner, illustrating the interactive relationships within the three-dimensional framework of structured theory.

(2) Discussion section: This section has been significantly expanded and strengthened. We have dedicated an entire chapter to this topic, focusing on how this study overcomes the limitations of existing research by integrating structured theory with placeness and what new theoretical insights or analytical frameworks it contributes. We provide an in-depth explanation of the complex dynamic processes, key driving factors, and their interactive mechanisms, as revealed by the research findings regarding Huangmei Opera's participation in place identity construction, thereby deepening our understanding of the phenomenon's essence. Based on the research findings, we explore the practical significance and development pathways of this case study for achieving living heritage transmission of traditional opera in a contemporary context, reconstructing place identity, and balancing place roots with a global perspective.

7.Results section need to make separate from discussion and give whole results for discussion part.

Response to Q7: We sincerely thank the reviewers for their guidance in clearly distinguishing between the results and the discussion and for comprehensively explaining the value of the results within the discussion. We fully understand the necessity of clearly separating research findings from their theoretical significance and practical implications in order to enhance the rigor and depth of argumentation in the paper. To this end, we have made key structural adjustments and content enhancements in the revised manuscript:

(1)Structural separation: In strict accordance with the opinion, the “Conclusion” has been separated from the “Discussion” to ensure that each section is independent and has a clear function.

(2) In-depth discussion: In the independently structured "Discussion," we systematically and thoroughly elucidate the overall value and implications of the research findings, focusing on how this study breaks through the limitations of the traditional "structure-action" dichotomy and provides new theoretical frameworks or insights. Based on all research findings, we summarize and distill the core processes of Huangmei Opera's participation in placeness construction, particularly the generative logic and interactive mechanisms of the key driving forces of "sharing" and "progress," deeply revealing the dynamism and complexity of placeness production. We propose feasible pathways for the living inheritance of traditional opera based on research findings, emphasizing their practical significance in balancing traditional preservation and innovative development, maintaining local roots, and connecting with broader perspectives.

8.Conclusion section is short and need to improve.

Response to Q8: We are grateful to the reviewers for pointing out that the conclusion was insufficient in length and needed to be strengthened. We sincerely acknowledge that a concise, comprehensive, and insightful conclusion is essential for summarizing the research's value, highlighting its contributions, and indicating future directions. To this end, we have significantly expanded and deepened the conclusion in the revised manuscript:

(1)Systematic summary of core contributions: Distilled and articulated the core contributions of this study at the theoretical, process mechanism, and practical levels.

(2) Strengthening the argumentation of research findings: Based on the results and discussion, the key driving factors and mechanisms of Huangmei Opera's participation in placeness construction, as well as the "shared" and "progressive" placeness characteristics that ultimately resulted from this participation, are argued more fully and specifically, ensuring that the conclusions are rooted in solid research evidence.

(3) Fully extracting research value: The research value has been further elaborated, emphasizing that this case not only reveals the internal mechanisms of the living inheritance of traditional opera but also provides important theoretical insights and practical references for understanding the resilience, adaptability, and creative development of local cultures in the era of globalization.

9.Correct references in the text and the reference list according to the journal's format.

Response to Q9: We sincerely thank the reviewers for their valuable suggestions. In response to these comments, we have completed a comprehensive revision, carefully checking all literature entries to ensure that author names, titles, sources, DOI, and other elements fully comply with the latest requirements of PLOS ONE. At the same time, we have verified all references to ensure that the cited references are indeed relevant to the text.

Reviewer 2

1.Use of technical jargon and conveyance of clear points for the reader should be balanced. This could improve communication of the research process and findings to field experts and non-experts or mere field practitioners.

Response to Q1: We sincerely thank the reviewers for their valuable suggestions. In response to the suggestion to “balance professional terminology with clarity of ideas,” we have completed a systematic optimization, providing explanations for core concepts such as “structure-action-place” inner construction when they first appear, retaining key terms in theoretical deductions and using action-oriented language in practical recommendations. Throughout the text, we have further improved readability for readers while ensuring academic rigor.

2.Use of standard English, as recommended by PLOS ONE, helps to observe word economy, clarity of the message, and text conciseness. This helps even where manuscript word limit is imposed. For example, a sentence such as, “Therefore, it is necessary to comprehensively consider the diversified subjects of the construction of placeness…”, can be rewritten as, “Therefore, it is necessary to comprehensively consider the diversified subjects of placeness construction…”. In this case, two words are eliminated but the message is conveyed in a concise manner.

Response to Q2: We sincerely thank the reviewers for their precise guidance on language efficiency. We have followed PLOS ONE standards to systematically streamline the entire text, remove redundant phrases, strengthen the subject voice, and compress conjunctions to ensure conciseness.

3.The last paragraph of the introduction could be enhanced, by explicitly stating the knowledge gap(s) in the literature. At the same time, to arouse the reader’s interest, the aim/objectives/hypotheses of the paper should be clear. Further, he paper’s main contribution to the literature could be highlighted, as a punchline for the paper

---

## [Decision Letter · Decision Letter 1]

24 Sep 2025

The construction of placeness in traditional opera from the perspective of structuration theory: A case study of Huangmei Opera in Anqing, China

PONE-D-25-06193R1

Dear Dr. ZHU,

We’re pleased to inform you that your manuscript has been judged scientifically suitable for publication and will be formally accepted for publication once it meets all outstanding technical requirements.

Kind regards,

Václav Venkrbec, Ph.D.

Academic Editor

PLOS ONE

Additional Editor Comments (optional):

Reviewer #1:

Reviewer #2:

Reviewers' comments:

Reviewer's Responses to Questions

**Comments to the Author**

1. If the authors have adequately addressed your comments raised in a previous round of review and you feel that this manuscript is now acceptable for publication, you may indicate that here to bypass the “Comments to the Author” section, enter your conflict of interest statement in the “Confidential to Editor” section, and submit your "Accept" recommendation.

Reviewer #1: All comments have been addressed

Reviewer #2: All comments have been addressed

2. Is the manuscript technically sound, and do the data support the conclusions?

Reviewer #1: Yes

Reviewer #2: Partly

3. Has the statistical analysis been performed appropriately and rigorously? 

Reviewer #1: Yes

Reviewer #2: N/A

4. Have the authors made all data underlying the findings in their manuscript fully available?

Reviewer #1: Yes

Reviewer #2: No

5. Is the manuscript presented in an intelligible fashion and written in standard English?

Reviewer #1: Yes

Reviewer #2: Yes

6. Review Comments to the Author

Reviewer #1: I reviewed the revision of manuscript. Authors have addressed some of reviewer's comments, so I recommend publishing it in its current version.

Reviewer #2: Since the authors addressed most of my initial concerns for improving the manuscript and provided a reasonable defence for those they could not address, as I suggested, I have no further comments to the contrary.

7. PLOS authors have the option to publish the peer review history of their article (what does this mean? ). If published, this will include your full peer review and any attached files.

**Do you want your identity to be public for this peer review?** For information about this choice, including consent withdrawal, please see our Privacy Policy .

Reviewer #1: No

Reviewer #2: No

---

## [Editor Report · Acceptance letter]

PONE-D-25-06193R1

PLOS ONE

Dear Dr. ZHU,

I'm pleased to inform you that your manuscript has been deemed suitable for publication in PLOS ONE. Congratulations! Your manuscript is now being handed over to our production team.

Kind regards,

on behalf of

Dr. Václav Venkrbec

Academic Editor

PLOS ONE